# Synthesis and Spectroscopic Characterization of Selected Phenothiazines and Phenazines Rationalized Based on DFT Calculation

**DOI:** 10.3390/molecules27217519

**Published:** 2022-11-04

**Authors:** Daniel Swoboda, Jacek E. Nycz, Nataliya Karaush-Karmazin, Boris Minaev, Maria Książek, Joachim Kusz, Radosław Podsiadły

**Affiliations:** 1Institute of Chemistry, Faculty of Science and Technology, University of Silesia in Katowice, ul. Szkolna 9, 40-007 Katowice, Poland; 2Department of Chemistry and Nanomaterials Science, Bohdan Khmelnytsky National University, 18031 Cherkasy, Ukraine; 3Institute of Physics, Faculty of Science and Technology, University of Silesia in Katowice, 75 Pułku Piechoty 1a, 41-500 Chorzów, Poland; 4Institute of Polymer and Dye Technology, Faculty of Chemistry, Lodz University of Technology, Stefanowskiego 12/16, 90-924 Lodz, Poland

**Keywords:** phenazine, phenothiazine, phosphorylation, heterocyclic, anomeric effect, phosphorescence, DFT studies

## Abstract

Two unique structures were isolated from the phosphorylation reaction of 10*H*-phenothiazine. The 5,5-dimethyl-2-(10*H*-phenothiazin-10-yl)-1,3,2-dioxaphosphinane 2-oxide (**2a**) illustrates the product of *N*-phosphorylation of phenothiazine. Moreover, a potential product of **2a** instability, a thiophosphoric acid **2b**, was successfully isolated and structurally characterized. Molecule **2a**, similarly to sulfoxide derivative **3**, possesses interesting phosphorescence properties due to the presence of d-pπ bonds. The X-ray, NMR, and DFT computational studies indicate that compound **2a** exhibits an anomeric effect. Additionally, the syntheses of selected symmetrical and unsymmetrical pyridine-embedded phenazines were elaborated. To compare the influence of phosphorus and sulfur atoms on the structural characteristics of 10*H*-phenothiazine derivatives, the high-quality crystals of (4a,12a-dihydro-12*H*-benzo[5,6][1,4]thiazino[2,3-*b*]quinoxalin-12-yl)(phenyl)methanone (**1**) and selected phenazines 5,12-diisopropyl-3,10-dimethyldipyrido[3,2-*a*:3′,2′-*h*]phenazine (**5**) and 5-isopropyl-*N,N*,3-trimethylpyrido[3,2-*a*]phenazin-10-amine (**6a**) were obtained. The structures of molecules **1**, **2a**, 2-mercapto-5,5-dimethyl-1,3,2-dioxaphosphinane 2-oxide (**2b**), 3,7-dinitro-10*H*-phenothiazine 5-oxide (**3**), **5** and **6a** were determined by single-crystal X-ray diffraction measurements.

## 1. Introduction

The phenothiazines and the phenazines are exceptional and versatile classes of electron-rich nitrogen heterocycles with broad pharmaceutical profiles and rich materials sciences applications [1,2,3]. A seemingly small difference between these types of molecules, i.e., the presence of a six-membered ring with nitrogen and sulfur atoms for phenothiazines and two nitrogen atoms for phenazines, manifests itself profoundly through their physicochemical and structural properties. The unique characteristics of both systems are present in (4a,12a-dihydro-12*H*-benzo[5,6][1,4]thiazino[2,3-*b*]quinoxalin-12-yl)(phenyl)methanone (**1**) and its analogs.

The phenazine class of *N*-heterocycles, numbering over 6000 compounds, is an important structural motif in functional materials, pharmaceuticals, agricultural, and natural products. Therefore, developing new efficient methods for preparing molecules with this privileged moiety is an essential endeavor in synthetic chemistry [4]. Their broad range of biological activity is thought to be due to their ability to undergo redox cycling in the presence of various reducing agents and molecular oxygen [5,6]. Phenazine natural products constitute biologically active metabolites produced by soil habitats and marine microorganisms [4]. The blue pigment pyocyanine, the fluorescent siderophore pyoverdine, phenazine-1-carboxylic acid, and the siderophore pyochelin are the most famous representatives. The biological properties of this class of natural products include antibiotic, antitumor, antimalarial, and antiparasitic activities [4]. They play an important role in microbial competitiveness, the suppression of soilborne plant pathogens, and virulence in human and animal hosts [5]. The described class of compounds is an intriguing topic of research.

Concerning the phenothiazines, many representatives of these electron-rich tricyclic nitrogen–sulfur heterocycles are the active components in various sedatives, tranquilizers, antiepileptics, antituberculosis, antipyretics, antitumor agents, bactericides, and parasiticides [7,8,9,10,11]. In addition to their physiological activities, the phenothiazines’ reversible oxidative properties give rise to deeply colored radical cations, making them attractive spectroscopic probes in molecular arrangements for photoinduced electron transfer studies and as scientific motif materials [12,13]. Phenothiazine derivatives can participate in reduction and oxidation processes manifested in their various oxidation states, such as neutral, cation radical, and oxygenated sulfoxide forms, providing synthetic perspectives, especially in pharmaceuticals [13,14,15]. Phenothiazines can act as antioxidants for many easily oxidizable substrates, including lubricants, rubber, polymers, and biological materials [16,17,18]. In particular, 10*H*-phenothiazine has been found to inhibit the autoxidation of methyl linoleate and phenoxazine to retard lipid peroxidation in rat brains [19]. It has also been suggested that the pharmacological activity of phenothiazines might be somehow related to their antioxidant or radical-trapping ability [20]. Another important application of phenothiazine and related compounds concerns their use as polymerization inhibitors to stop the reactions of radical polymerization, which may occur during the preparation and workup of acrylic monomers or during their storage [21,22]. *N*-Alkylation of phenothiazine has led to pharmaceutically essential substances, such as promethazine (antihistaminic), ethopropazine (anti-Parkinson), and dimethoxanate (antitussive). The discovery of the antipsychotic agent chlorpromazine in the early 1950s and the appearance of even more potent phenothiazine psychopharmacological agents constitute a landmark event in the history of the medical and psychiatric sciences. As a result of their numerous medicinal and industrial applications, extensive derivatization and/or functionalization of these parent structures continue unabated.

The current study presents the syntheses and spectroscopic characterization of selected phenothiazines and phenazines rationalized based on density functional theory (DFT) calculations.

## 2. Results and Discussion

### 2.1. Synthesis and Structural Characterization of Studied Compounds

The *N*-benzoylated, *N*-alkylated, or *N*-arylated phenothiazines and related compounds are stable molecules [23,24,25,26,27]. Some of their representatives are highly potent inhibitors of cancer cell growth and tubulin polymerization [26]. The summary of molecular structures of compounds discussed in this paper is presented in Figure 1.

Keglevich and co-workers obtained the *N*-phosphorylated phenothiazine, i.e., *O*,*O*-diethyl (10*H*-phenothiazine-10-yl)phosphonate. The authors claimed that the phosphorylated phenothiazine was not stable and decomposed. They reported difficulties during its separation from the other components, especially the *N*-ethylated phenothiazine, i.e., 10-ethyl-10*H*-phenothiazine. The authors realized that the distribution profile and product yields strongly depend on the used base, solvents, and heating mode (conventional or microwave irradiation). They determined the product’s composition by LC-MS techniques. The highest *O*,*O*-diethyl (10*H*-phenothiazine-10-yl)phosphonate yield was observed in the case of using NaH as a base in refluxing DMF (52%), and lower values were observed for refluxing toluene (15%). Under other reaction conditions, the authors did not identify the *N*-phosphorylated product. The results obtained from phosphorylation reactions were opposite to *N*-ethylation, where the highest yields were observed for applying K_2_CO_3_ under microwave irradiation conditions (80%), and the lowest yields were observed for NaH and toluene with conventional heating (5%) [25]. Inspired by the work of Prof. Keglevich’s team, we performed a similar type of reaction; however, as a phosphorylating agent, we chose 2-chloro-5,5-dimethyl-1,3,2-dioxaphosphinane 2-oxide. This electrophile is known for its high tendency to crystallize (Figure 2) [28,29]. In order to synthesize the 5,5-dimethyl-2-(10*H*-phenothiazin-10-yl)-1,3,2-dioxaphosphinane 2-oxide (**2a**), we performed a reaction between the in situ generated sodium salt of phenothiazine and 2-chloro-5,5-dimethyl-1,3,2-dioxaphosphinane 2-oxide in THF. The synthetic route with the structures of reagents is presented in Figure 2. Product **2a** was obtained with a good 58% yield. The unexpected side-product 2-mercapto-5,5-dimethyl-1,3,2-dioxaphosphinane 2-oxide (**2b**) was only isolated to prove its presence (Figure 2). The radical rearrangement of product **2a** may explain the formation of molecule **2b** through the disintegration of the P-N bond and the establishment of a new P-S bond. To elucidate the possible origin of side-product and the relationship between the molecular structures of stable acyl or aryl phenothiazine derivatives, high-quality single crystals of (4a,12a-dihydro-12*H*-benzo[5,6][1,4]thiazino[2,3-*b*]quinoxalin-12-yl)(phenyl)methanone (**1**) and compound **2a** were cultivated by slow evaporation from a hot DMSO or a hot mixture of THF and hexane, respectively.

The phenothiazine molecule possesses a unique non-planar butterfly structure. Tian and co-workers drew attention to the derivatives having an S=O (or SO_2_) and P=O group, which contains the d-pπ bond that could hinder a possible intense intermolecular π-π stacking, which is beneficial for inhibiting excited triplet-triplet annihilation (TTA) [27]. Tian et al. presented some examples of molecules with a S=O bond (in the SO_2_ group), which shows a sharp increase in TTA by 19 times compared to that without this group.

Compound **1** crystallizes in the monoclinic system with the *P*2_1_/*n* space group (Figure 1) and demonstrates layered packing similar to the phenothiazine model system described by Tian and co-workers [27]. Intermolecular π-π stacking distances vary in the range of 3.670-3.736 Å (Figure 1c). Additionally, a rigid 2D network structure of **1** is stabilized via intermolecular C-H⋯N interactions with distances of 2.690 Å. The bending angles in the phenothiazine core of molecule **1** were defined as the ∠N12−S11−C10 and ∠N5−N6−C7, as shown in Figure 1a,b, and were found to be 130.88(5)° and 131.92(8)°. The distance between the centroids of the phenyl and azine rings is equal to 3.712(1)Å.

The X-ray bond lengths and bond angles for molecule **1** are provided in detail in Appendix A. The computational DFT results confirm a bent-core structure of compound **1** and are consistent with a crystal structure in bending angles and bond length presented in Appendix A. The calculated bending angles (135.4° and 135.8°, respectively) are close to X-ray data, indicating that the bending structure is not primarily caused by lattice packing but is due to the steric hindrance inherent in phenothiazines.

The intermolecular interactions between atoms in the crystal of molecule **1** were also studied using Hirshfeld surfaces analysis [30,31] via the *d_norm_*, shape index, and curvedness (Figure 1d,e). The surfaces are shown as transparent to visualize the molecular skeleton. The *d_norm_* surface exhibits two intensive red spots, which correspond to short intermolecular CH…N interactions with distances of 2.690 Å (Figure 1d). The presence of π-π stacking interactions is indicated by red and blue triangles on the shape-index surface (brown ellipses in Figure 1e) and by a flat region on the curvedness (Figure 1f). A network of intermolecular CH…N and π-π stacking interactions hinder molecular movement in crystal **1**, which helps to reduce possible energy losses through non-radiative relaxation channels.

Molecules **2a** crystallize to a monoclinic system, *P*2_1_/*n*, with unit cell dimensions of a = 8.8248(3) Å, b = 16.5758(4) Å, c = 11.8216(4) Å, V = 1678.23(9) Å^3^, and Z = 4; the monoclinic angle (β) is 103.951(3)°, and the unit cell contains 4 molecules (Figure 2a–d). The ∠N10−P1−O1 and ∠O2−P1−O3 in compound **2a** were found to be 103.67° and 112.26°, respectively (Figure 2b). The bending angle ∠C1−S9−C8 in the phenothiazine core of *N*-phosphorylated phenothiazine **2a** is smaller than in molecule **1a** and was found to be 126.87°. The DFT calculations of the phenothiazine derivative **2a** structure by the B3LYP/6-31G(d,p) method indicate a minimal lattice packing effect in the dioxaphosphinane fragment in terms of ∠N10−P1−O1 (102.96°) and ∠O2−P1−O3 (113.70°) and bond lengths presented in Appendix A. The calculated bending angle in the phenothiazine core (∠C1−S9−C8) of molecule **2a** is 131.17°. The bending angles in the middle phenothiazine ring are denoted as Θ_S_ and Θ_N_ and were calculated to be 98.30° (exp.: 97.90°) and 117.13° (exp.: 115.60°) (Figure 2e). The Hirshfeld *d_norm_* surfaces of **2a** indicate the stabilization of crystal packing with numerous CH…O and CH…S intermolecular interactions (Figure 2f,g). The shape index visualizes the complementarity of molecules in the crystal and shows that there are no π-π stacking interactions in the volume (Figure 2h,i).

The thiophosphoric acid of > P(=S)OH type could exist in one of two tautomeric structures, either with P=S and P-OH bonds or with P=O and P-SH. The second tautomeric form of molecule **2b**, which we suggest, is much less proposed in the chemical literature (Figure 2). The P=O (P1–O1 1.446(4) Å) and P-S (P1–S1B 2.0006(10) Å, P1–S1A 2.0006(10) Å) bond lengths in molecule **2b** (Appendix A) are in good agreement with those reported for a similar type of compounds [32,33,34]. Molecules **2b** crystallize into an orthorhombic system, *P*bca, with unit cell dimensions of a = 11.1587(17) Å, b = 11.8846(13) Å, c = 13.1114(16) Å, V = 1738.8(4) Å^3^, and Z = 8; the orthorhombic angle (β) is 90°, and the unit cell contains 8 molecules. The -SH group is disordered over two positions with a refined occupancy ratio of 0.49(5):0.51(5).

Compound **3** crystallizes to a monoclinic system, *P*2_1_/*n*, with unit cell dimensions of a = 8.2753(3) Å, b = 19.9331(7) Å, c = 8.5875(3) Å, V= 1398.32(9) Å^3^, and Z = 4; the monoclinic angle (β) is 99.197(3)°, and the unit cell contains 4 molecules (Figure 3c). The oxygen atom from the -SO group is disordered over two positions with a refined occupancy ratio of 0.949(5):0.051(5). Therefore, Hirshfeld surfaces analysis is not applicable for **3**. Both molecules **2a** and **3** crystallize in the monoclinic system and have no π-π interactions (for **2a**) or are weak (for **3**, Figure 3b). This can be explained by a reduction in the electron density on the benzene rings on both sides of phenothiazine upon the *N*-phosphorylation for compound **2a** or the oxidation of sulfur (electron-donating group) to sulfoxides (electron-withdrawing group) for molecule **3**. Experimental bending angles in the middle phenothiazine ring are 98.31°for Θ_S_ and 124.47° for Θ_N_ (Figure 3d). We can find some similarities in physicochemical properties among our molecule **3** and sulfonyl derivatives published by Tian et al. [27]. They also crystallize into the monoclinic system and have abundant intermolecular C-H⋯π and C-H⋯O interactions in the crystal. The multiple intermolecular interactions help to suppress molecular motions and thereby reduce the possible energy loss via non-radiative relaxation channels.

In the final step, we decided to replace heteroatoms of the phenothiazine core to establish the influence of such modification on the structural properties of studied compounds. According to the literature, changing the nitrogen atom into a sulfur atom, which leads to thianthrene derivatives, would preserve the bent spatial arrangement of the atoms, hindering π-π stacking interactions. Such a configuration is abundant in molecules already described in previous paragraphs. Aiming to increase the number of π-stacks in the crystal structure, we chose more planar phenazines as synthetic targets, where nitrogen is placed instead of a sulfur atom. There are already published examples of synthesizing this class of compounds in the literature. Therefore, further experiments were conducted to explore some synthetic routes that gave the aforementioned compounds. Additionally, we tried to explain the mechanism of producing symmetrical and unsymmetrical phenazines.

We have recently reported the direct oxidative condensation to introduce the carbon-nitrogen bond by a one-step method [35]. To synthesize the symmetrical phenazine **5** and the unsymmetrical phenazines **6**, we applied two other types of chemical transformations. One led to symmetrical molecules and was based on the oxidative condensation of two identical molecules, i.e., 8-*iso*propyl-2-methylquinolin-5-amine (**4a**) (Figure 3). In the second route, we chose the oxidative condensation between two electron-rich amines (ERA) *N*^1^,*N*^1^-dimethylbenzene-1,4-diamine and selected 8-(alkyl)-2-methylquinolin-5-amines **4** (Figure 4). We chose silver carbonate adsorbed on celite as an oxidizing agent in both methods. The transformation occurred with a good yield for symmetrical phenazine **5** (Figure 3) and a poor yield for the unsymmetrical phenazines **6** (Figure 4). The deposition of metallic silver during the reaction progress suggests the involvement of electron transfer processes from ERA to silver(I) salt. This assumption is grounded on the preparation of the 1,2,4,5-tetrakis(dimethylamino)benzene dication iodide-triiodide through iodination of 1,2,4,5-tetrakis(dimethylamino)benzene patented by Staab et al. [36].

Seth and co-workers reported the synthesis of symmetrical phenazines on non-radical C-H activation mode of C-N coupling during the oxidative self-coupling of anilines catalyzed by Pd-Ag and Cu-Ag nanoclusters [37,38]. They show the superiority of Pd-Ag over Cu-Ag nanoclusters during the synthesis of phenazines, with yields up to 87%. As another product, they isolated azoaryls with good yields (up to 89%) by applying the same nanoclusters, with superiority of Cu-Ag. In order to discern the character of the reaction mechanism, ionic or radical, they used radical scavenger (2,2,6,6-tetramethylpiperidin-1-yl)oxyl (TEMPO). Due to the lack of evidence of the impact of TEMPO, they excluded the radical mechanism of phenazine formation. Inspired by the work of Seth and co-workers [37,38], we employed only silver carbonate as a mildly basic oxidizing agent. We observed a similar double C-N bond formation involving two different amine molecules. However, the synthesis of unsymmetrical phenazines **6** is more complicated. Both used amine molecules **4** and *N^1^,N^1^*-dimethylbenzene-1,4-diamine could participate in electron transfer to silver carbonate, however, at a different rate, due to their redox potentials (Figure 5). EPR techniques do not readily detect free radicals **4a**’ derived from ERA [39,40]. The evidence of their presence may be an azoarene of the **7** types (Figure 5), the formation of which accompanied the synthesis of phenazines, similar to Seth and co-workers’ studies [37,38]. For any synthesis of unsymmetrical phenazine **6**, we identified compounds **6e** and **7**, which were confirmed by the original sample addition. This observation partially explains the much lower yields of unsymmetrical phenazines **6** (Figure 4). On the basis of the literature and our present results, we propose a plausible mechanism for the formation of phenazines (symmetrical **5** and unsymmetrical **6**) under applied reaction conditions. For simplicity, the postulated pathways are based on amine **4e** (Figure 5). The presented mechanism illustrates the influence of the silver(I) cation as an oxidizing agent and oxygen from the air. During transformation, molecules of water are created.

Compound **5** crystallizes in the tetragonal system (*P*4_2_/n space group) with a = 21.1688(8) Å, b = 21.1688(8) Å, c = 4.6776(3) Å, and α = β = γ = 90°. Figure 4 presents the molecular (a) and crystal (b) structure of phenazine **5**. Figure 4a–c shows that the planar molecular skeleton of molecule **5** promotes the formation of π-stacks with intermolecular distances of 3.652–3.690 Å. Hirshfeld’s *d_norm_* surface indicates these regions as white or light blue colors in Figure 4d. More clearly, these π-stacking arrangements are identified in the shape index map with red and blue triangles (brown ellipses in Figure 4e) and the curvedness map as large green flat areas along the aromatic structural fragments of compound **5** (Figure 4f).

Compound **6a** crystallizes in the triclinic system (*P-*1 space group) with a = 9.0731(6) Å, b = 9.3169(7) Å, and c = 11.5962(8) Å. Figure 5 presents the molecular (a) and packing (b) structure of phenazine **6a**. The distance between the centroids of the azine rings of neighboring molecules is equal to 3.628(1) Å.

The Hirshfeld *d_norm_* surface for molecule **6a** illustrates the prominent red spot, which corresponds to the intermolecular CH⋯N interaction with a distance of 2.892 Å (Figure 5b,c). The intermolecular H…H and C⋯H contacts (Figure 5b,c) appear as pale red areas in the *d_norm_*. The latter belongs to the C-H⋯π interactions. The shape index (Figure 5d) confirms the presence of π-π stacking interactions as closely spaced red and blue triangles encircled by brown ellipses. Figure 5e shows π-π stacking with a flat zone in a curvedness map.

Mortzfeld et al. [41] reported the alkylation methodology of phenazine. As alkylation reagents, they chose methyl iodide, trimethyloxonium tetrafluoroborate, and dimethyl sulfate. They received stable mono *N*-alkylated products with good to poor yield: 86% for dimethyl sulfate, 25% for trimethyloxonium tetrafluoroborate, and only 19% for methyl iodine

### 2.2. Anomeric Effect of Compound ***2a***

The fundamental impact of stereochemical properties on a molecule’s biological activity makes the conformational aspect of research essential in many fields of organic chemistry. One of the crucial phenomena altering conformational behavior and, consequently, the reactivity of saturated heterocyclic systems is the anomeric effect, also known as the Edward–Lemieux effect [42,43]. It characterizes the propensity of heteroatomic substituents neighboring a heteroatom embedded in the cyclohexane ring to favor the axial position instead of the less sterically hindered equatorial orientation. In the IUPAC Compendium of Chemical Terminology, this is “a special case of a general preference for synclinal (gauche) conformations about the bond C-Y in the system X-C-Y-C where X and Y are heteroatoms having non-bonding electron pairs, commonly at least one of which is nitrogen, oxygen or fluorine” [44].

In the 1970s, Bentrude et al. initiated conformation studies concerning saturated 1,3,2-dioxaphosphinane rings [45,46], which structures mirror features of the carbohydrate molecules. As inferred, the preferential axial position of substituents is the one adjacent to oxygen. Hence, the anomeric effect was determined to be the most dominant conformation-controlling factor for compounds with 1,3,2-dioxaphosphinane rings.

Our results show that compound **2a** also undergoes the anomeric effect. Both forms of chair conformers were conveniently identified by ^31^P{^1^H}-NMR spectroscopy and X-ray diffraction measurements (Figure 2, Figure 6 and Figure 7). For molecule **2a**, the non-bonding p oxygen orbitals interact with phosphorus, and the charge is transferred through the nitrogen atom to the thiazine-type ring, which possesses empty π-accepting orbitals. Moreover, the energies of the two possible chair conformers differ by about 1.45 kcal/mol or 6.07 kJ/mol (gas-phase B3LYP/6-31g(d,p) calculation, Figure 6), which makes the existence of an equilibrium between these two forms at room temperature possible. As shown in the ^31^P{^1^H}-NMR spectrum (Figure 7), compound **2a** has two separated conformers, δ_1_ = −6.91 and δ_2_ = −7.02 ppm. The relative signal intensity of both molecules **2a** is in a ratio (ca. 72:1).

The nature of the anomeric effect is still a matter of controversy, and it is interesting to note that this challenging situation has attracted the attention of chemists and physicists for over half a century. It has been shown that steric, electrostatic, and stereoelectronic factors are all important forces that influence the anomeric effect. In the literature, there are several proposed explanations for the validity of the anomeric effect. One could be a stabilizing hyperconjugation contribution between the unshared electron pair on the oxygen atom in a 1,3,2-dioxaphosphinane ring and the σ* orbital for the P-C bond in the axial position (Figure 6). For conformer 2, the stabilizing hyperconjugation effect is weaker. Therefore, this form is less stable and was not isolated but only identified in the ^31^P{^1^H}-NMR spectrum (Figure 7).

The occurrence of the anomeric phenomenon entails certain geometric features that are consistent with the above-noted hyperconjugative delocalization: some bonds are lengthened, others are shortened, and the angles are widened or reduced [47]. The assignments of conformation were based on NMR, X-ray, literature analysis, and our previous results [28,32,45]. According to the Karplus equation and the couplings constant, the order of chemical shifts for CH_2_ protons, δ_Haxial_ > δ_Hequatorial_, was determined. The methylene protons of compound **2a** showed two distinct double doublets in the ^1^H-NMR spectrum at δ = 3.60 and 3.74 ppm. The expected long-range couplings (^3^*J*_P,Haxial_ = 3.7 Hz and ^3^*J*_P,Hequatorial_ = 22.9 Hz) between phosphorus and one of the bridged methylene protons are observed.

### 2.3. Photophysical Properties of the Phenothiazine Derivatives ***2a*** and ***3***

Figure 8a depicts the absorption spectra of molecule **2a** in MeCN in the 600–190 nm region. Phenothiazine **2a** has two absorption bands located at 280 and 234 nm, which can be ascribed to the n-π* and π-π* transitions, respectively, typical of many phenothiazines [48]. In addition, we can observe a band below 200 nm, similar to other phenothiazines [49]. In comparison with 10*H*-phenothiazine (λ_max_ = 252 nm and 316 nm [50]), the presence of *N*-phosphoryl substituent in compound **2a** caused a hypsochromic effect. Non-planar structure of **2a** makes the n-π assignment questionable.

According to quantum chemical calculations by the TDDFT/B3LYP/6-31G(d,p) method, the weak absorption in the range 300–275 nm (absorption band 280 nm in Figure 8) in the experimental spectrum of molecule **2a** is manifested as a broadening on the right side of the spectrum and corresponds to a series of low-intensity transitions S_0_ → S_1_, S_0_ → S_2_ and S_0_ → S_3_ at 286 (*f* = 0.0148), 273 (*f* = 0.0141), and 265 nm (*f* = 0.0401), respectively. The electronic transitions and frontier molecular orbitals (HOMO—highest occupied molecular orbital; LUMO—lowest unoccupied molecular orbital) are presented in Appendix A and Figure 8b, Appendix A. All these three transitions into S_1_, S_2_, and S_3_ states of compound **2a** possess a similar orbital nature, S_0_ → S_1_ (HOMO → LUMO), S_0_ → S_2_ (HOMO → LUMO + 1), and S_0_ → S_3_ (HOMO → LUMO + 2, HOMO-1→ LUMO), and are due to the charge transfer from the phenothiazine core to benzene fragments (Figure 8b). The same orbital nature of the first absorption band is also characteristic of *N*-methyl phenothiazine [48]. The next intense absorption band (234 nm) was calculated in the range of 250–220 nm with a maximum at 238 nm (Appendix A), and it can be assigned to the electronic transitions into the S_4_, S_5_, and S_6_ excited states (Appendix A).

The fluorescent band of compound **2a** (Figure 9) is located at 374 nm, and it is also blue-shifted in comparison with the emission band of 10*H*-phenothiazine (λ_em_ = 454 nm [50]) and *N*-methyl-phenothiazine (λ_em_ = 450 nm [51]). The TDDFT calculations for molecule **2a** clearly indicate the observed emission band at 374 nm (Figure 9), which corresponds to the vertical S_1_→S_0_ transition calculated at 386 nm (Appendix A).

We also investigated the emission behavior of compound **2a** at 77 K (Figure 10). When excited at 291 nm, molecule **2a** shows an emission band with two peaks at 434 nm and 457 nm. Its profile of phosphorescence is similar to the phosphorescence spectra of pure 10*H*-phenothiazine (λ 500 nm [50], λ 535 nm [52]). Similar to absorption and fluorescence, it is clear that the phosphoryl moiety in compound **2a** causes a large hypsochromic effect (c.a. 70 nm) in the phosphorescence spectrum. Phosphorescence bands of molecule **2a** are located in the same region as the phosphorescence of two *N*-acylphenothiazine derivatives, namely 10*H*-phenothiazin-10-yl)(phenyl)methanone and 1-(10*H*-phenothiazin-10-yl)ethan-1-one) [27].

We also recorded the phosphorescence spectrum of compound **2a** in crystals at 298 K (Figure 11). Molecule **2a** exhibits a relatively low-intensity fluorescence peak at 373 nm and an intense phosphorescence peak centered at 540 nm. Phosphorescence characteristics of *N*-phosphorylated phenothiazine **2a** are similar to phosphorescence spectra of 10*H*-phenothiazin-10-yl)(phenyl)methanone and 1-(10*H*-phenothiazin-10-yl)ethan-1-one) that have phosphorescence emission at 460–600 nm, with the maximum located at 530 nm and 525 nm, respectively [8]. The TDDFT results for compound **2a** show that the lowest triplet (T_1_) state lies at 2.46 eV and involves mixed electronic configurations HOMO → LUMO, HOMO-1 → LUMO + 2, HOMO-2→ LUMO + 3, HOMO-3 → LUMO + 1. Similar to other phenothiazines [52,53], for molecule **2a**, S_1_, S_2_, S_3_, and T_1_ possess charge transfer characters with a certain degree of locally excited nature. Thus, a thermally activated reverse singlet-to-triplet intersystem crossing (RISC) can probably occur from the upper triplet to singlet levels providing phosphorescence in compound **2a** since the singlet-triplet (S_1_-T_1_) energy gap is large (0.76 eV).

The absence of π-π stacking contacts in molecule **2a** and strong charge-transfer character of the T_1_ state (Figure 8b) provide efficient σ-π orbital mixing, which is responsible for the increase in spin-orbit coupling and intersystem crossing in this compound affecting high phosphorescence [54].

Figure 12 shows the steady-state emission spectra of the phenothiazine derivative **2a** under different conditions.

Dinitrophenothiazine *S*-oxide **3** has four absorption bands located at 271 nm, 359 nm, 398 nm, and 528 nm (Figure 13a). In comparison with phenothiazine *S*-oxide (λ_max_ 229 nm, 271 nm, 303 nm, 336 nm [55]), the two additional nitro groups presented in the phenothiazine *S*-oxide skeleton caused a large bathochromic effect. TDDFT calculations predict that the first high-intensity absorption maximum in molecule **3** (Figure 13) belongs to the S_0_→S_1_ transition calculated at 376 nm (*f* = 0.6989) in DMSO solution (Appendix A). The S_0_→S_1_ transition is provided by the HOMO→LUMO configuration and possesses a charge transfer (CT) nature; this is mainly CT from sulfur to nitro groups (Figure 13b). For comparison, the first S_0_→S_1_ transition for phenothiazine *S*-oxide was calculated at 305 nm, which confirms the presence of a bathochromic effect due to nitro groups. The difference in the calculated and experimental values of the wavelengths can be explained by an incomplete allowance for solvation effects within the framework of the TDDFT/B3LYP/6-31G(d,p) level of theory. Figure 13b indicates that the HOMO is an antibonding orbital with respect to C-N and C-S bonds in the central ring, showing that there is a strong C-C bonding character, whereas the LUMO is mostly a non-bonding orbital in the center. Thus, the HOMO-LUMO excitation leads to much shorter and stronger C-N and C-S bonds in the S_1_ excited state, but C-C bonds are becoming weaker (Appendix A).

Compound **3** shows a fluorescent band located at 647 nm (Figure 14). The phenothiazine *S*-oxide or its *N*-alkyl analogs exhibit fluorescence in the 350–450 nm region [56,57]. For molecule **3**, TDDFT calculations indicate the observed emission band at 647 nm as the S_1_ → S_0_ transition calculated at 568 nm (Appendix A). Two electron-withdrawing nitro groups remove the electron density from the sulfur atom and decrease the LUMO energy, and the emission band of dinitrophenothiazine *S*-oxide is red-shifted by 200 nm. Moreover, molecule **3** shows a large Stokes shift (122 nm), which indicates that the geometry of the singlet excited state differs from the geometry of the ground state. In the S_0_ state of molecule **3**, the bending angles in the middle phenothiazine ring denoted as Θ_S_ and Θ_N_ were calculated to be 95.08° and 122.78°, respectively, while in S_1_, these angles are 104.50 and 127.27 (Appendix A). Much larger distortions are calculated in the C-N and C-S bond lengths upon the S_1_ → S_0_ transition (Appendix A) according to the strong differences in the bonding-nonbonding characters of the frontier orbitals (Figure 13b). Typical distortions also occur in the nitro groups since LUMO is bonding for the C-N links and anti-bonding for the N-O chemical bonds.

In contrast to molecule **2a** and 10*H*-phenothiazine, the 5,5-dioxide-based push-pull chromophore [27] compound **3** shows no room-temperature phosphorescence, which is typical for nitroaromatic compounds. Asymmetric NO_2_ stretching and hindered rotation of nitro-groups represent accepting and promoting modes, respectively, for efficient phosphorescence quenching of compound **3**.

## 3. Materials and Methods

### 3.1. Materials

All experiments were carried out in an atmosphere of dry argon, and flasks were flame dried. Solvents were dried by usual methods (diphenyl ether, diethyl ether, and THF over benzophenoneketyl, CHCl_3_, and CH_2_Cl_2_ over P_4_O_10_, hexane, and pyridine over sodium-potassium alloy) and distilled. Chromatographic purification was carried out on silica gel 60 (0.15–0.3 mm, Macherey-Nagel GmbH & Co. KG, Dueren, Germany). Crotonaldehyde, 4a,12a-dihydro-12*H*-benzo[5,6][1,4]thiazino[2,3-*b*]quinoxaline, *N^1^,N^1^*-dimethylbenzene-1,4-diamine, 2,2-dimethylpropane-1,3-diol, 10*H*-phenothiazine, phosphoryl chloride, 2-*iso*propylaniline, isoquinolin-5-amine, quinolin-5-amine, sodium hydride (dry, 95%), stannous chloride, silver carbonate on Celite and *o*-toluidine were purchased from Sigma–Aldrich (Poznań, Poland), and were used without further purification.

### 3.2. Instrumentation

NMR spectra were obtained with Avance 400 and 500 spectrometers (Bruker, Billerica, MA, USA) operating at 500.2 or 400.2 MHz (^1^H) and 125.8 or 100.6 MHz (^13^C) and 202.47 (^31^P) at 21 °C. Chemical shifts referenced to ext. TMS (tetramethylsilane) (^1^H, ^13^C) or 85% H_3_PO_4_ (^31^P), or using the residual CHCl_3_ signal (δ_H_ 7.26 ppm) and CDCl_3_ (δ_C_ 77.1 ppm) as internal references for ^1^H and ^13^C-NMR, respectively. Coupling constants are given in Hz. The LCMS-IT-TOF analysis was performed on an Agilent 1200 Series binary LC system coupled to a micrOTOF-Q system mass spectrometer (BrukerDaltonics, Bremen, Germany). High-resolution mass spectrometry (HRMS) measurements were performed using a Synapt G2-Si mass spectrometer (Waters, New Castle, DE, USA) equipped with an ESI source and quadrupole-time-of-flight mass analyzer. To ensure accurate mass measurements, data were collected in centroid mode, and mass was corrected during acquisition using leucine enkephalin solution as an external reference (Lock-Spray) (Waters, New Castle, DE, USA). The measurement results were processed using the MassLynx 4.1 software (Waters, Milford, MA, USA) incorporated within the instrument. A Nicolet iS50 FTIR spectrometer was used for recording spectra in the IR range of 4000–400 cm^−1^. FTIR spectra were recorded on a Perkin Elmer (Schwerzenbach, Switzerland) spectrophotometer in the spectral range of 4000–450 cm^−1^ with the samples in the form of KBr pellets. Elementary analysis was performed using a Vario EL III apparatus (Elementar, Langenselbold, Germany). Differential scanning calorimetry (DSC) measurements were performed using a Q2000 calorimeter (TA Instruments, New Castle, DE, USA) in a nitrogen stream at a scanning rate of 10 °C/min. Samples were analyzed in aluminum pans in the temperature range of 50 to 350 °C. Melting points were determined on an MPA100 OptiMelt melting point apparatus (Stanford Research Systems, Sunnyvale, CA, USA) and are uncorrected. UV/Vis absorption spectra were measured on a SHIMADZU UV-VIS-NIR spectrophotometer using quartz (Suprasil) cuvettes (10 mm path length). Fluorescence spectra were recorded on an FLS-900 spectrofluorimeter (Edinburgh Instruments, Edinburgh, UK) and were measured in quartz (Suprasil) cuvettes (10 mm path length) at 25 °C. The phosphorescence spectrum was recorded on an FLS-900 spectrofluorimeter (Edinburgh Instruments, Edinburgh, UK) at 77 K using a 3 mm (inner diameter) quartz tube inside a quartz liquid nitrogen Dewar flask.

### 3.3. Synthesis of (4a,12a-Dihydro-12H-benzo[5,6][1,4]thiazino[2,3-b]quinoxalin-12-yl)(phenyl)methanone Followed Our Procedure Described in the Literature [23]

**(4a,12a-dihydro-12*H*-benzo[5,6][1,4]thiazino[2,3-*b*]quinoxalin-12-yl)(phenyl)methanone** (**1**) [23], m.p. = 200.0–200.5 °C; ^1^H-NMR (DMSO-d_6_; 400.2 MHz) δ = 7.25 (t, ^3^*J*_H,H_ = 7.4 Hz, 2H, aromatic), 7.31 (t, ^3^*J*_H,H_ = 7.0 Hz, 1H, aromatic), 7.41 (d, ^3^*J*_H,H_ = 7.7 Hz, 3H, aromatic), 7.50 (t, ^3^*J*_H,H_ = 8.8 Hz, 2H, aromatic), 7.64 (t, ^3^*J*_H,H_ = 7.7 Hz, 1H, aromatic), 7.69 (d, ^3^*J*_H,H_ = 7.8 Hz, 1H, aromatic), 7.73 (t, ^3^*J*_H,H_ = 7.7 Hz, 1H, aromatic), 7.95 (d, ^3^*J*_H,H_ = 8.3 Hz, 1H, aromatic), 8.11 (d, ^3^*J*_H,H_ = 8.1 Hz, 1H, aromatic); ^13^C{^1^H}-NMR (DMSO-d_6_; 100.6 MHz) δ = 124.8, 125.1, 127.2, 127.5, 127.7, 127.9, 128.0, 128.1, 128.4, 130.2, 130.3, 130.9, 134.8, 135.9, 138.6, 139.7, 145.3, 149.8, 169.5; CCDC (The Cambridge Crystallographic Data Centre) 2177507.

### 3.4. Phosphorylation of Phenothiazine

The synthesis of 2-chloro-5,5-dimethyl-1,3,2-dioxaphosphinane 2-oxide followed our procedure described in the literature [28]. The synthesis of 5,5-dimethyl-2-(10*H*-phenothiazin-10-yl)-1,3,2-dioxaphosphinane 2-oxide (**2a**) was based on procedures described in the literature [58].

To a suspension of NaH (0.96 g, 40.0 mmol) in THF (100 mL), 10*H*-phenothiazine (4.00 g, 20.1 mmol) was added and stirred until the evolution of H_2_ ceased. Reagents were stirred under reflux for 30 min. under argon. 2-Chloro-5,5-dimethyl-1,3,2-dioxaphosphinane 2-oxide (3.70 g, 20.1 mmol) was then added to the reaction mixture, which was refluxed overnight. After evaporation of the solvent to give a solid, water (20 mL) and chloroform (100 mL) were added. The organic layer was separated, and the aqueous layer was extracted four times with chloroform. The combined organic layers were dried over MgSO_4_. After the solvent evaporated, the crude product was purified by column chromatography on silica gel using methanol/dichloromethane as eluent to afford a crude solid, and finally, crystallization from a mixture of CH_2_Cl_2_ and hexane occurred to yield solids as follows:

**5,5-Dimethyl-2-(10*H*-phenothiazin-10-yl)-1,3,2-dioxaphosphinane 2-oxide** (**2a**) 0.66 g (2.78 mmol, 58 %), white powder, m.p. = 183–185 °C; ^1^H-NMR (CDCl_3_; 500.2 MHz) δ = 0.74 (s, 3H, CH_3_), 1.25 (s, 3H, CH_3_), 3.60 (ddd, ^3^*J*_P,H_ = 22.9 Hz, ^3^*J*_H,H_ = 11.2 Hz, ^4^*J*_H,H_ = 1.5 Hz, 2H, C*H*), 3.74 (dd, ^3^*J*_H,H_ = 10.5 Hz, ^3^*J*_P,H_ = 3.7 Hz, 2H, C*H*), 7.17 (td, ^3^*J*_H,H_ = 7.6 Hz, ^4^*J*_H,H_ = 1.2 Hz, 2H, aromatic), 7.26 (dd, ^3^*J*_H,H_ = 7.7 Hz, ^4^*J*_H,H_ = 1.5 Hz, 2H, aromatic), 7.34 (dd, ^3^*J*_H,H_ = 7.7 Hz, ^4^*J*_H,H_ = 1.4 Hz, 2H, aromatic), 7.60 (dd, ^3^*J*_H,H_ = 8.0 Hz, ^4^*J*_H,H_ = 0.9 Hz, 2H, aromatic); ^13^C{^1^H}-NMR (CDCl_3_; 125.8 MHz) δ = 20.5, 22.0, 32.1 (d, ^3^*J*_P,C_ = 32.1 Hz), 77.5 (d, ^3^*J*_P,C_ = 6.9 Hz), 123.6, 126.0, 126.8, 127.7, 127.6, 130.0 (d, ^3^*J*_P,C_ = 5.9 Hz), 139.5 (d, ^3^*J*_P,C_ = 6.9 Hz); ^31^P-NMR (CDCl_3_; 202.5 MHz) δ_1_ = −6.91 (t, ^3^*J*_P,H_ = 22.7 Hz), ^31^P{^1^H}-NMR (CDCl_3_; 202.47 MHz) δ_1_ = −6.91 and δ_2_ = −7.02; GC-MS: t_r_ = 9.6 min, (EI) M^+^ = 347.1 (68%), (M-C_5_H_10_O_3_P)^+^ = 198.1 (100); UV-Vis (metanol; λ [nm] (logε)): 273 (3.12), 252 (3.58), 230 (4.01), 204(3.99); IR (KBr): ν = 3059.24, 2984.65, 2969.43, 2886.37, 1463.03, 1298.53, 1269.99, 1225.79, 1055.73, 1008.36, 995.08, 981.81, 841.68, 795.30, 773.30, 759.03, 509.49, 552.50 cm^−1^; CCDC (The Cambridge Crystallographic Data Centre) 2177508. 2-Mercapto-5,5-dimethyl-1,3,2-dioxaphosphinane 2-oxide (**2b**) CCDC (The Cambridge Crystallographic Data Centre) 2178722.

### 3.5. Synthesis of 3,7-Dinitro-10H-phenothiazine 5-oxide (***3***)

10*H*-Phenothiazine (10.0 g, 50.2 mmol) was added in one portion to 100 mL of 65% nitric acid stirred in a wide beaker at room temperature. The immediate evolution of brown vapor was observed, and the reaction mixture’s temperature increased to 45 °C. After stirring for 1h, the deep red solution was filtered through a sintered glass funnel under reduced pressure, and the obtained solid was flushed with 200 mL of water and 100 mL of methanol. The crude product was boiled for 5 min in 300 mL of THF, filtered, and dried under reduced pressure. To obtain the crystals, the saturated solution of the purified product in CHCl_3_ was kept at −20 °C for a week.

**3,7-Dinitro-10*H*-phenothiazine 5-oxide** (**3**) [24] 10.5 g (34.4 mmol, 68.5 %) as a yellow powder, mp_dec._ = 310.0-316.5 °C; ^1^H-NMR (DMSO-d_6_; 500.2 MHz) δ = 7.61 (d, ^3^*J*_H,H_ = 9.1 Hz, 2H, aromatic), 8.48 (dd, ^3^*J*_H,H_ = 9.1 Hz, ^4^*J*_H,H_ = 2.6 Hz, 2H, aromatic), 8.96 (d, ^4^*J*_H,H_ = 2.6 Hz, 1H, aromatic), 12.32 (s, 1H, NH); ^1^H-NMR (DMSO-d_6_/KOD; 500.2 MHz) δ = 7.56 (d, ^3^*J*_H,H_ = 9.1 Hz, 2H, aromatic), 8.42 (dd, ^3^*J*_H,H_ = 9.1 Hz, ^4^*J*_H,H_ = 2.7 Hz, 2H, aromatic), 8.89 (d, ^4^*J*_H,H_ = 2.6 Hz, 1H, aromatic); ^13^C{^1^H}-NMR (DMSO-d_6_; 100.6 MHz) δ = 118.8, 121.5, 128.1, 128.2, 139.9, 141.7; ^13^C{^1^H}-NMR(DMSO-d_6_/KOD; 100.6 MHz) δ = 120.2, 121.2, 127.7, 128.4, 140.9, 141.9; CCDC (The Cambridge Crystallographic Data Centre) 2177509.

### 3.6. General Procedure for the Synthesis of 8-(Alkyl)-2-methylquinolin-5-amines

The synthesis of 2,8-dimethylquinoline, 8-(*iso*propyl)-2-methylquinolin-5-amine (**4a**) and 2,8-dimethylquinolin-5-amine (**4b**) followed our procedure described in the literature [35].

Synthesis of 2,8-dimethylquinoline.

Toluene (50 mL) and crotonaldehyde (2.6 mL, 2.2 g, 31.4 mmol) were added to the solution of *o*-toluidine (1.7 g, 15.7 mmol) in aqueous 6 M HCl (200 mL) and were heated under reflux for 16 h. The mixture was allowed to cool down to room temperature. The aqueous layer was separated and neutralized with an aqueous solution of K_2_CO_3_. After extraction with CH_2_Cl_2_ (3 × 50 mL), the organic layer was separated and dried over MgSO_4_, then filtered and distilled bp 100–110 °C/3 mmHg. The liquid mixture was dissolved in concentrated 36% HCl (100 mL) at 5 °C, and ZnCl_2_ (2.7 g, 20.0 mmol) was added with vigorous stirring for one hour. The precipitate was filtered and washed with cold 3 M aq. HCl and dried in air. The solid was washed with *i*PrOH and dried. The received white solid was added to a 10% ammonia solution and extraction with Et_2_O (3 × 50 mL). The organic layer was separated and dried over MgSO_4_ to afford:

**2,8-Dimethylquinoline** as a colorless oil, 1.8 g (11.6 mmol, 74%), bp 98–100 °C/3 mmHg; ^1^H-NMR (CDCl_3_; 400.2 MHz) δ = 2.74 (s, 3H, CH_3_), 2.80 (s, 3H, CH_3_), 7.24 (d, ^3^*J*_H,H_ = 8.3 Hz, 1H, aromatic), 7.34 (dd, ^3^*J*_H,H_ = 8.1 Hz, ^3^*J*_H,H_ = 7.0 Hz, 1H, aromatic), 7.50 (d, ^3^*J*_H,H_ = 7.0 Hz, 1H, aromatic), 7.58 (d, ^3^*J*_H,H_ = 8.0 Hz, 1H, aromatic), 7.97 (d, ^3^*J*_H,H_ = 8.3 Hz, 1H, aromatic); ^13^C{^1^H}-NMR (CDCl_3_; 125.8 MHz) δ = 18.0, 25.6, 26.9, 121.6, 125.2, 125.5, 126.3, 129.4, 136.2, 136.5, 147.0, 157.8.

Synthesis of 2,8-dimethyl-5-nitroquinoline.

2,8-Dimethylquinoline (1.2 g, 7.5 mmol) was dissolved in a mixture of concentrated H_2_SO_4_ and HNO_3_ (4.5 and 10.5 mL, respectively) at 5 °C. After stirring for 1 h at room temperature, no evolution of gas was observed, so the reaction mixture was heated up to 70 °C and stirred overnight. After this time, the reaction mixture was poured down into a beaker containing 25 g of ice and 25 mL of water, and the precipitated solid was filtered off, washed with 10 mL of cold water, and dried on air, giving:

**2,8-Dimethyl-5-nitroquinoline** as a beige solid, 1.1 g (5.6 mmol, 74%); DSC = 78.80–78.86 °C; ^1^H-NMR (CDCl_3_; 500.2 MHz) δ = 2.78 (s, 3H, CH_3_), 2.87 (d, ^4^*J*_H,H_ = 1.0 Hz, 3H, CH_3_), 7.49 (d, ^3^*J*_H,H_ = 8.9 Hz, 1H, aromatic), 7.59 (dd, ^3^*J*_H,H_ = 7.9 Hz, ^4^*J*_H,H_ = 1.0 Hz, 1H, aromatic), 8.22 (d, ^3^*J*_H,H_ = 7.9 Hz, 1H, aromatic), 8.91 (d, ^3^*J*_H,H_ = 8.9 Hz, 1H, aromatic); ^13^C{^1^H}-NMR (CDCl_3_; 125.8 MHz) δ = 19.0, 25.4, 119.3, 123.6, 124.6, 127.3, 132.0, 143.8, 145.6, 146.6, 159.2; HRMS (TOF-ES+): *m*/*z* Calcd for C_11_H_11_N_2_O_2_ (M + H)^+^ = 203.0821, Found 203.0818.

Synthesis of 2,8-dimethylquinolin-5-amine (**4b**).

Stannous chloride (47.4 g, 250.0 mmol) was added to a stirred solution of 2,8-dimethyl-5-nitroquinoline (5.0 g, 25.0 mmol) in 6M hydrochloric acid (100 mL) and methanol (300 mL). After stirring for 30 min at room temperature, the reaction mixture was brought to reflux and stirred for 3 h. After cooling to room temperature, the mixture was basified with aqueous ammonia and extracted with chloroform (3 × 50 mL). The combined extract was dried over MgSO_4_ and evaporated to afford:

**2,8-Dimethylquinolin-5-amine** (**4b**) as a beige solid, 3.5 g (20.2 mmol, 81%); DSC = 62.89–63.54 °C; ^1^H-NMR (CDCl_3_; 500.2 MHz) δ = 2.67 (d, ^4^*J*_H,H_ = 1.0 Hz, 3H, CH_3_), 2.73 (s, 3H, CH_3_), 3.97 (bs, 2H, NH_2_), 6.66 (d, ^3^*J*_H,H_ = 7.5 Hz, 1H, aromatic), 7.21 (d, ^3^*J*_H,H_ = 8.6 Hz, 1H, aromatic), 7.30 (dd, ^3^*J*_H,H_ = 7.6 Hz, ^4^*J*_H,H_ = 1.0 Hz, 1H, aromatic), 8.03 (d, ^3^*J*_H,H_ = 8.6 Hz, 1H, aromatic); ^13^C{^1^H}-NMR (CDCl_3_; 125.8 MHz) δ = 17.5, 25.5, 109.3, 117.2, 120.1, 127.0, 129.66, 129.67, 140.1, 147.4, 157.6; HRMS (TOF-ES^+^): *m*/*z* Calcd for C_11_H_11_N_2_O_2_ (M + H)^+^ = 173.1079, Found 173.1074.

### 3.7. Synthesis of 5,12-Diisopropyl-3,10-dimethyldipyrido[3,2-a:3′,2′-h]phenazine (***5***)

The solution of 8-*iso*propyl-2-methylquinolin-5-amine (**4a**) (400 mg, 2.00 mmol) in toluene (50 mL) was brought to gentle reflux, and silver carbonate on celite (2.5 g, 50% *w*/*w*, 4.50 mmol Ag_2_CO_3_) was added in small portions over 30 min. The reaction mixture was stirred under reflux for a further 4 h and filtered under reduced pressure without cooling. The solid residue was flushed with ca. 50 mL of ethyl acetate until a clean, colorless filtrate was observed. The obtained solution was left to stand in the refrigerator overnight. The next day, the formed light yellow precipitate was collected by filtration, flushed with warm methanol, and recrystallized from chloroform to afford yellow crystals:

**5,12-Diisopropyl-3,10-dimethyldipyrido[3,2-*a*:3**′,**2**′**-*h*]phenazine** (**5**) 224 mg (0.57 mmol, 57.0%); DSC = 289.70 °C; ^1^H-NMR (CDCl_3_; 500.2 MHz) δ = 1.54 (2d, ^3^*J*_H,H_ = 6.9 Hz, 12H, CH(C*H*_3_)_2_), 2.85 (s, 6H, CH_3_), 4.44 (2 septet, ^3^*J*_H,H_ = 6.9 Hz, 2H, C*H*(CH_3_)_2_), 7.58 (2d, ^3^*J*_H,H_ = 8.2 Hz, 2H, aromatic), 8.21 (2s, 2H, aromatic), 9.57 (2d, ^3^*J*_H,H_ = 8.3 Hz, 2H, aromatic); ^13^C{^1^H}-NMR (CDCl_3_; 125.8 MHz) δ = 23.3, 25.7, 28.0, 122.4, 124.2, 125.5, 133.2, 140.6, 141.3, 148.2, 151.7, 160.1; IR (KBr):ν = 2956, 1586, 1472, 1420, 1373, 1233, 1136, 1050, 885, 847, 807cm^−1^; HRMS (TOF-ES+): *m*/*z* Calcd for C_26_H_27_N_4_ (M + H)^+^ = 395.2236, Found 395.2230; CCDC (The Cambridge Crystallographic Data Centre) 2177510.

### 3.8. Synthesis of Unsymmetrical Phenazines (***6a***–***d***)

The solution of appropriate quinoline (**4a**–**c**) or isoquinoline **4d** (2.00 mmol) and *N^1^,N^1^*-dimethylbenzene-1,4-diamine (286 mg, 2.10 mmol) in toluene (50 mL) was brought to a gentle reflux, and silver carbonate on celite (2.5 g, 50% *w*/*w*, 4.50 mmol Ag_2_CO_3_) was added in small portions over 30 min. The reaction mixture was stirred under reflux for a further 4 h and filtered under reduced pressure without cooling. The solid residue was flushed with ca. 50 mL of ethyl acetate until a clean, colorless filtrate was observed. The obtained solution was concentrated and purified by column chromatography on silica gel using chloroform and/or ethyl acetate/hexane eluent system to afford:

**5-Isopropyl-*N,N*,3-trimethylpyrido[3,2-*a*]phenazin-10-amine** (**6a**) as orange crystals, 50 mg (0.15 mmol, 7.5%); DSC = 209.35 °C; ^1^H-NMR (CDCl_3_; 500.2 MHz) δ = 1.47 (d, ^3^*J*_H,H_ = 6.9 Hz, 6H, 2CH_3_), 2.82 (s, 3H, C*H*_3_), 3.22 (s, 6H, NC*H*_3_), 4.33 (septet, ^3^*J*_H,H_ = 6.9 Hz, 1H, C*H*(CH_3_)_2_), 7.23 (d, ^3^*J*_H,H_ = 2.8 Hz, 1H, aromatic), 7.51 (d, ^3^*J*_H,H_ = 8.3 Hz, 1H, aromatic), 7.57 (dd, ^3^*J*_H,H_ = 9.5 Hz, ^4^*J*_H,H_ = 2.8 Hz, 1H, aromatic), 8.00 (d, ^4^*J*_H,H_ = 0.8 Hz, 1H, aromatic), 8.08 (d,^3^*J*_H,H_ = 9.5 Hz, 1H, aromatic), 9.48 (d, ^3^*J*_H,H_ = 8.2 Hz, 1H, aromatic); ^13^C{^1^H}-NMR (CDCl_3_; 125.8 MHz) δ = 23.3, 25.6, 27.7, 40.7, 104.3, 121.5, 122.0, 124.1, 125.6, 129.8, 133.3, 138,2, 140.1, 141.7, 144.1, 148.7, 149.4, 151.0, 159.9; IR (KBr): ν =3872, 3650, 2954, 2922, 2864, 2801, 2366, 1734, 1628, 1601, 1497, 1461, 1407, 1356, 1313, 1225, 1186, 1145, 1081, 1011, 966, 916, 884, 851, 810 cm^−1^; HRMS (TOF-ES^+^): *m*/*z* Calcd for C_17_H_15_N_4_ (M + H)^+^ = 331.1923, Found 331.1919; CCDC (The Cambridge Crystallographic Data Centre) 2177511.

***N*,*N*,3,5-Tetramethylpyrido[3,2-*a*]phenazin-10-amine** (**6b**) as a brown powder, 33 mg (0.11 mmol, 5.5%); DSC = 217.68 °C; ^1^H-NMR (CDCl_3_; 500.2 MHz) δ = 2.83 (s, 3H, C*H*_3_), 2.87 (d, ^4^*J*_H,H_ = 1.2 Hz, 3H, C*H*_3_), 3.23 (s, 6H, NC*H*_3_), 7.23 (d, ^3^*J*_H,H_ = 2.8 Hz, 1H, aromatic), 7.52 (d, ^3^*J*_H,H_ = 8.2 Hz, 1H, aromatic), 7.57 (dd, ^3^*J*_H,H_ = 9.5 Hz, ^3^*J*_H,H_ = 2.8 Hz, 1H, aromatic), 7.98 (dd, ^4^*J*_H,H_ = 1.3 Hz,1H, aromatic), 8.08 (d, ^3^*J*_H,H_ = 9.5 Hz, 1H, aromatic), 9.48 (d, ^3^*J*_H,H_ = 8.2 Hz, 1H, aromatic); ^13^C{^1^H}-NMR (CDCl_3_; 125.8 MHz) δ = 19.0, 25.6, 40.7, 104.3, 121.5, 122.1, 124.0, 129.4, 129.9, 133.2, 138.2, 139.3, 140.0, 142.0, 144.1, 149.8, 151.1, 160.2; IR (KBr): ν = 3788, 3435, 2918, 1632, 1608, 1505, 1468, 1437, 1405, 1358, 1313, 1262, 1188, 1151, 1087, 1151, 1087, 1014, 914, 869, 803 cm^−1^; HRMS (TOF-ES^+^): *m*/*z* calcd for C_19_H_19_N_4_ (M + H)^+^ = 303.1610, Found 303.1606.

***N*,*N*-Dimethylpyrido[3,2-*a*]phenazin-10-amine** (**6c**) as a red powder, 32 mg (0.12 mmol, 6.0%); DSC = 189.46 °C; ^1^H-NMR (CDCl_3_; 500.2 MHz) δ = 3.25 (s, 6H, NC*H*_3_), 7.24 (d, ^3^*J*_H,H_ = 2.5 Hz, 1H, aromatic), 7.62 (ddd, ^3^*J*_H,H_ = 9.5 Hz, ^4^*J*_H,H_ = 2.8 Hz, ^4^*J*_H,H_ = 1.1 Hz, 1H, aromatic), 8.11 (d,^3^*J*_H,H_ = 4.3 Hz, 1H, aromatic), 8.13 (d, ^3^*J*_H,H_ = 4.4 Hz, 1H, aromatic), 8.18 (d, ^3^*J*_H,H_ = 9.5 Hz,1H, aromatic), 9.06 (dd, ^3^*J*_H,H_ = 4.5 Hz, ^3^*J*_H,H_ = 1.8 Hz, 1H, aromatic), 9.63 (dd, ^3^*J*_H,H_ = 8.2 Hz, ^4^*J*_H,H_ = 1.1 Hz, 1H, aromatic); ^13^C{^1^H}-NMR (CDCl_3_; 125.8 MHz) δ = 40.5, 103.7, 121.9, 122.0, 126.7, 130.0, 130.6, 131.4, 133.3, 138.4, 139.6, 141.9, 144.4, 149.9, 151.29, 151.34; IR (KBr): ν = 3340, 2923, 1630, 1601, 1509, 1472, 1413, 1358, 1261, 1149, 1096, 1024, 918, 808 cm^−1^; HRMS (TOF-ES^+^): *m*/*z* Calcd for C_17_H_15_N_4_ (M + H)^+^ = 275.1297, found 275.1290.

***N*,*N*-Dimethylpyrido[4,3-*a*]phenazin-10-amine** (**6d**) as a red powder, 20 mg (0.07 mmol, 3.5%); DSC = 204.68 °C; ^1^H-NMR (CDCl_3_; 500.2 MHz) δ = 3.26 (s, 6H, NC*H*_3_), 7.25 (d, ^3^*J*_H,H_ = 2.9 Hz, 1H, aromatic), 7.65 (dd, ^3^*J*_H,H_ = 9.5 Hz, ^4^*J*_H,H_ = 2.8 Hz, 1H, aromatic), 7.93 (dd, ^3^*J*_H,H_ = 9.2 Hz, ^4^*J*_H,H_ = 0.7 Hz, 1H, aromatic), 8.03 (d, ^3^*J*_H,H_ = 9.2 Hz, 1H, aromatic), 8.12 (d, ^3^*J*_H,H_ = 9.5 Hz,1H, aromatic), 8.89 (d, ^3^*J*_H,H_ = 5.5 Hz, 1H, aromatic), 9.10 (dt, ^3^*J*_H,H_ = 5.5 Hz, ^4^*J*_H,H_ = 0.8 Hz, 1H, aromatic), 9.28 (d, ^4^*J*_H,H_ = 1.0 Hz, 1H, aromatic); ^13^C{^1^H}-NMR (CDCl_3_; 125.8 MHz) δ = 40.7, 103.9, 118.2, 122.8, 126.9, 128.4, 129.4, 130.0, 136.4, 139.3, 141.16, 141.25, 144.7, 146.1, 150.5, 151.5; IR (KBr): ν = 3427, 2962, 1627, 1501, 1414, 1356, 1262, 1219, 1097, 1028, 805, 703 cm^−1^;HRMS (TOF-ES^+^): *m*/*z* Calcd for C_17_H_15_N_4_ (M + H)^+^ = 275.1297, found 275.1296.

### 3.9. Synthesis of ***6e*** and ***7***

The solution of *N^1^,N^1^*-dimethylbenzene-1,4-diamine (286 mg, 2.10 mmol) in toluene (50 mL) was brought to a gentle reflux, and silver carbonate on celite (2.5 g, 50% *w*/*w*, 4.50 mmol Ag_2_CO_3_) was added in small portions over 30 min. The reaction mixture was stirred under reflux for 4 h and filtered under reduced pressure without cooling. The solid residue was flushed with ca. 50 mL of ethyl acetate until a clean, colorless filtrate was observed. The obtained solution was concentrated and purified by column chromatography on silica gel using chloroform and/or ethyl acetate/hexane eluent system to afford:

***N*^2^,*N*^2^,*N*^7^,*N*^7^-tetramethylphenazine-2,7-diamine** (**6e**) as a red powder, 90 mg (0.34 mmol, 33.8%); DSC = 255.69–255.86 °C; ^1^H-NMR (CDCl_3_; 500.2 MHz) δ = 3.15 (s, 12H, NC*H*_3_), 7.07 (d, ^3^*J*_H,H_ = 2.8 Hz, 2H, aromatic), 7.52 (dd, ^3^*J*_H,H_ = 9.6 Hz, ^4^*J*_H,H_ = 2.8 Hz, 2H, aromatic), 7.93 (dd, ^3^*J*_H,H_ = 9.5 Hz, 2H, aromatic); ^13^C{^1^H}-NMR (CDCl_3_; 125.8 MHz) δ = 40.8, 104.3, 122.5, 129.1, 139.6, 143.2, 149.8; IR (KBr): ν = 2956, 1586, 1472, 1421, 1372 cm^−1^; HRMS (TOF-ES^+^): *m*/*z* Calcd for C_16_H_19_N_4_ (M + H)^+^ = 267.1610, found 267.1609.

**(*E*)-4,4′-(diazene-1,2-diyl)bis(*N*,*N*-dimethylaniline)** (**7**) as a brown powder, 92 mg (0.34 mmol, 34.3%); DSC = 257.68 °C [59]; ^1^H-NMR (CDCl_3_; 500.2 MHz) δ = 3.06 (s, 12H, NC*H*_3_), 6.76 (d, ^3^*J*_H,H_ = 9.1 Hz, 4H, aromatic), 7.80 (d, ^3^*J*_H,H_ = 9.0 Hz, 4H, aromatic); ^13^C{^1^H}-NMR (CDCl_3_; 100.6 MHz) δ = 29.9, 40.6, 111.9, 124.1, 144.3, 151.7.

### 3.10. Hirshfeld Surface Calculations

Hirshfeld surfaces were calculated using Crystal Explorer 21.5 [31] to analyze the close contacts in the phenothiazine and phenazine compounds. The normalized contact distance (*d_norm_*) values are mapped with red-blue-white colors, where the closer contacts with *d_norm_* < 0 are represented in red areas, while the blue region indicates longer contacts with *d_norm_* > 0, and white areas correspond to contacts with distances equal to the van der Waals radii (*d_norm_* = 0). The shape index visualizes complementary molecules in the crystal with red (hollows) and blue (bumps) colors, where molecular surfaces touch one another. Curvedness maps show planar regions on the surface in green, which is separated by blue edges corresponding to large positive curvature.

### 3.11. Density Functional Theory (DFT) Computations

The molecular structures of the phenothiazines and phenazines in the ground singlet state (S_0_) were optimized with the DFT/B3LYP/6-31G(d,p) approach [60,61,62,63]. For the calculated molecules, all vibrational wavenumbers were found to be real, indicating the location of the true minimum on the hypersurface of the total energy. The first excited singlet (S_1_) and triplet (T_1_) states, as well as vertical S_0_-S_n_ transitions, were calculated within the time-dependent (TD) DFT approximation [64] with the same B3LYP/6-31G(d,p) method [61,62,63]. The solvent effects on the absorption and fluorescence spectra were taken into account using the polarized-continuum model [65]. The Gaussian 16 program package [66] was used for all the DFT and TDDFT calculations.

### 3.12. X-ray Diffraction Experiments

The data for **1**, **2a**, **3**, **5**, and **6a** were collected using a SuperNova diffractometer (Agilent Technologies, currently Rigaku Oxford Diffraction). Accurate cell parameters were determined and refined using the CrysAlis^Pro^ program [67]. Additionally, integration of the collected data was performed with this program. The data for **2b** were collected using an Xcaliburdiffractometer (Oxford Diffraction, currently Rigaku Oxford Diffraction). Determination and refinement of the cell parameters, as well as the integration of the collected data, were performed using the CrysAlis^Pro^ program [68]. All the structures were solved using direct methods with the SHELXS97 program and then refined using the SHELXL-2018/3 program [69]. Nonhydrogen atoms were refined with anisotropic displacement parameters. The hydrogen atoms were fixed at calculated distances and allowed to ride on the parent atoms.

## 4. Conclusions

A series of *N*-heteroaromatics have been strategically designed and synthesized to compare their spectroscopic properties and rationalize them on the basis of theoretical calculations (Figure 1). We described and analyzed the crystal structure of the *N*-benzoylated phenothiazine-phenazine hybrid molecule **1**. Compounds of this type possess rich future chemistry potential for photoluminescent materials [27] and polymerization photoinitiating systems [23]. The presented synthesis protocols allowed obtaining the symmetrical phenazines with good yields (molecules **5** and **6e**) and with low yields for unsymmetrical representatives **6a**–**d**, giving six novel compounds. The crystal structures of phenazines **5** and **6a** were determined by single-crystal X-ray diffraction measurements. Interestingly, molecules **6a**–**d** are structurally related to the well-known biological stain neutral red, the analogs of which exhibit promising photosensitizing [70] and electrochemical properties [71].

Our results show that phosphorylation of phenothiazine proceeds exclusively to one new product **2a**, with a newly formed P-N bond together with rearrangement leading to a sulfurylated thiophosphoric acid **2b**. The S_1_ and T_1_ excited states of compound **2a** possess hybridized local and charge-transfer characters and show excellent luminescent properties. This benefit makes molecule **2a** a potentially highly luminous molecule. The *N*-phosphoryl moiety does not participate directly in the low-energy excitations of molecule **2a** (it does not contribute to frontier molecular orbitals, Figure 8b) but provides a hypsochromic shift. According to the energy gap law [72], this blue shift leads to hindered non-radiative quenching of the T_1_ state and to sustainable phosphorescence.

Comparing the molecular orbitals involved in excitations for **2a** and **3**, it should be noted that for both phenothiazines, these MOs are localized in the phenothiazine ring, but their shapes and charge transfer nature differ significantly. The presence of phosphorus in molecule **2a** provides a strong stabilizing hyperconjugation effect, and the charge is transferred through the nitrogen atom to the thiazine-type ring. Additionally, the absence of π–π stacking interactions in molecule **2a** and the strong charge-transfer character of the T_1_ state provide efficient σ-π orbital mixing responsible for the increase in spin-orbit coupling and intersystem crossing in this compound, which contributes to high phosphorescence. Phenothiazine **3** has no stabilizing hyperconjugation effect; the first S_0_→S_1_ transition possesses a charge transfer character mainly from sulfur to nitro groups. In contrast to molecule **2a**, compound **3** does not exhibit phosphorescence at room temperature, which is typical for nitroaromatic compounds.

## Data Availability

The data set presented in this study is available in this article.

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
