# Peer review of "Synthesis and Spectroscopic Characterization of Selected Phenothiazines and Phenazines Rationalized Based on DFT Calculation"

_molecules, 2022, doi:10.3390/molecules27217519_

Round 1

Reviewer 1 Report

The manuscript molecules-1978911 "Synthesis and Spectroscopic Characterization of Selected Phenothiazines and Phenazines Rationalized Based on DFT calculation" by Swoboda et al. describes the synthesis of phenothiazine and phenazine derivatives and the study of their photophysical properties. The authors have interesting results, so I think that this paper will be of interest to the readers of Molecules.

Questions and comments:

1) The authors should add information about the proposal and implementation of this study at the end of the Introduction. Therefore, the first sentence of Section 2.1 can be moved to the Introduction.

2) I misunderstood the choice of compounds for this study. If there are any supposed structural patterns, then write it clearly. I also recommend writing more clearly the novelty of this study, especially when compared with previous ones.

3) It is very unclear in the manuscript which compounds were synthesized earlier and which compounds were synthesized for the first time. Therefore, I do not understand why synthetic schemes, conditions and yields of previously obtained compounds are given and discussed. The same applies to the experimental part and supplementary materials, i.e. if the compound was obtained earlier, it is not necessary to duplicate their spectral characteristics.

4) The chemical structures of all used compounds must be presented in graphical form (for example, make one figure with all structural formulas).

5) The quality of many figures should be improved.

6) The Introduction uses materials based on old publications, i.e. only 13 references out of 66 are studies of the last 5 years. So, I recommend the authors to strengthen the Introduction part about synthesis and applications of phenothiazine and phenazine derivatives. Recent review articles of 2022 on this topic should be added. For example, 10.3390/molecules27010276, 10.1039/D2TC02085H, 10.1016/j.dyepig.2022.110806.

Author Response

Dear Editor,

We want to thank the referees whose comments helped us improve the manuscript's clarity. Further, we provide detailed comments on the referees’ notes. All changes made in the text are highlighted in green color.

 A point-by-point response to comments from Reviewer 1

The manuscript molecules-1978911 "Synthesis and Spectroscopic Characterization of Selected Phenothiazines and Phenazines Rationalized Based on DFT calculation" by Swoboda et al. describes the synthesis of phenothiazine and phenazine derivatives and the study of their photophysical properties. The authors have interesting results, so I think that this paper will be of interest to the readers of Molecules.

Questions and comments:

1) The authors should add information about the proposal and implementation of this study at the end of the Introduction. Therefore, the first sentence of Section 2.1 can be moved to the Introduction.

Answer: Many thanks for this important recommendation. We have included the proposal and implementation of this study at the end of the Introduction.

2) I misunderstood the choice of compounds for this study. If there are any supposed structural patterns, then write it clearly. I also recommend writing more clearly the novelty of this study, especially when compared with previous ones.

Answer: Many thanks for this important crochet info. We added an explanation.

3) It is very unclear in the manuscript which compounds were synthesized earlier and which compounds were synthesized for the first time. Therefore, I do not understand why synthetic schemes, conditions and yields of previously obtained compounds are given and discussed. The same applies to the experimental part and supplementary materials, i.e. if the compound was obtained earlier, it is not necessary to duplicate their spectral characteristics.

Answer: Molecules: 8-(isopropyl)-2-methylquinoline, 12-phenyl-4a,12a-dihydro-12H-benzo[5,6][1,4]thiazino[2,3-b]quinoxaline (1b) and 4a,12a-dihydro-12H-benzo[5,6][1,4]thiazino[2,3-b]quinoxaline (1c), 8-(isopropyl)-2-methyl-5-nitroquinoline, 8-(isopropyl)-2-methylquinolin-5-amine (4a) and have been deleted. We left the spectroscopic characteristics of compound previously described by our team namely (4a,12a-dihydro-12H-benzo[5,6][1,4]thiazino[2,3-b]quinoxalin-12-yl)(phenyl)methanone (1). In the present work, the characteristic is more detailed. The 2,8-dimethylquinoline was obtained through our new methodology. The 2,8-dimethyl-5-nitroquinoline and 2,8-dimethylquinolin-5-amine (4b) are new products. We elaborated a new procedure of synthesis of known: 2,8-dimethylquinoline, 3,7-dinitro-10H-phenothiazine 5-oxide (3). We left the spectroscopic characteristics of this compound to show that both procedures lead to the same molecule.

4) The chemical structures of all used compounds must be presented in graphical form (for example, make one figure with all structural formulas).

Answer: The suggested correction has been made. In the present version of our manuscript, all presented molecules are presented in Scheme 5 in the Conclusion part.

5) The quality of many figures should be improved.

Answer: The suggested correction has been made.

6) The Introduction uses materials based on old publications, i.e. only 13 references out of 66 are studies of the last 5 years. So, I recommend the authors to strengthen the Introduction part about synthesis and applications of phenothiazine and phenazine derivatives. Recent review articles of 2022 on this topic should be added. For example, 10.3390/molecules27010276, 10.1039/D2TC02085H, 10.1016/j.dyepig.2022.110806.

Answer: The suggested correction has been made. The recent review articles have been added, ref. Nr. 1-3.

Some improvements in the writing have been made. I have carefully revised the whole manuscript and tried to avoid grammar or syntax errors. Besides, I have asked several skilled authors of English language papers to check the English. Thank you so much for your help. I appreciate it.

Yours sincerely, Jacek Nycz (on behalf of all co-Authors)

Reviewer 2 Report

The authors present the structural and spectroscopic properties of diverse phenothiazine and phenazine based compounds. The subject is of interest for the journal readers, and the paper presents some valuable data. I suggest the publication after the consideration of the following major points:

1.       Although the paper contains a numerous information and a large number of the compounds, I struggle to find out the relations between the chosen structures.  Sometimes, it seems like sole listing of the compounds. For example, X-ray data of compounds are listed but no intercorrelations or relations between structures are discussed (in abstract: To compare the influence of  phosphorus and sulfur atom on the structural characteristics of 10H-phenothiazine derivatives.... but no comparison in the text is found).  Please, make clear justification for choosing diverse phenothiazines and phenazines with different functionalities

2.       Authors are strongly advised to give all molecular structures of the interest on one Scheme. The structures and their names are hard to follow in a presented form of the manuscript.

3.       Characterization of the molecules 1b and 1c is redundant since no information on these molecules is presented in the manuscript. Also, the characterization of the compounds known in literature is unnecessary and repetitive

4.        Furthermore, as the paper refers to the spectroscopic characterization of phenazine and phenothiazine derivatives, it should also include photophysical properties of the phenazine derivatives (5, 6  and 7) 

Author Response

Dear Editor,

We want to thank the referees whose comments helped us improve the manuscript's clarity. Further, we provide detailed comments on the referees’ notes. All changes made in the text are highlighted in green color.

 A point-by-point response to comments from Reviewer 2

The authors present the structural and spectroscopic properties of diverse phenothiazine and phenazine based compounds. The subject is of interest for the journal readers, and the paper presents some valuable data. I suggest the publication after the consideration of the following major points:

  1. Although the paper contains a numerous information and a large number of the compounds, I struggle to find out the relations between the chosen structures. Sometimes, it seems like sole listing of the compounds. For example, X-ray data of compounds are listed but no intercorrelations or relations between structures are discussed (in abstract: To compare the influence of phosphorus and sulfur atom on the structural characteristics of 10H-phenothiazine derivatives.... but no comparison in the text is found). Please, make clear justification for choosing diverse phenothiazines and phenazines with different functionalities

Answer: Many thanks for this important crochet info. We added an explanation. In particular, we added a comparison of the influence of phosphorus and sulfur atoms and the association of crystal packing with photoluminescent properties for phenothiazines in Part 2.3 and Conclusion.

The N-phosphoryl moiety does not participate directly in the low-energy excitations of molecule 2a (it does not contribute to frontier molecular orbitals, Figure 8b) but provides a hypsochromic shift. According to the energy gap law, this blue shift leads to hindered non-radiative quenching of the T1 state and to sustainable phosphorescence.

Comparing the molecular orbitals involved in excitations for 2a and 3, it should be noted that for both phenothiazines, these MOs are localized in the phenothiazine ring, but their shapes and charge transfer nature differ significantly. The presence of phosphorus in the 2a molecule provides a strong stabilizing hyperconjugation effect, and the charge is transferred through the nitrogen atom to the thiazine-type ring. Additionally, the absence of π–π stacking interactions in 2a and the strong charge-transfer character of the T1 state provide efficient σ-π orbital mixing responsible for the increase of spin-orbit coupling and intersystem crossing in this compound, which contributes to high phosphorescence. Phenothiazine 3 has no stabilizing hyperconjugation effect; the first S0→S1 transition possesses a charge transfer character mainly from sulfur to nitro groups. In contrast to 2a, compound 3 does not exhibit phosphorescence at room temperature, which is typical for nitroaromatic compounds. Asymmetric NO2 stretching and hindered rotation of nitro-groups represent accepting and promoting modes, respectively, for efficient phosphorescence quenching of compound 3.

  1. Authors are strongly advised to give all molecular structures of the interest on one Scheme. The structures and their names are hard to follow in a presented form of the manuscript.

Answer: The suggested correction has been made. In the present version of our manuscript, all presented molecules are presented in Scheme 5 in the Conclusion part.

  1. Characterization of the molecules 1b and 1c is redundant since no information on these molecules is presented in the manuscript. Also, the characterization of the compounds known in literature is unnecessary and repetitive

Answer: Molecules: 8-(isopropyl)-2-methylquinoline, 12-phenyl-4a,12a-dihydro-12H-benzo[5,6][1,4]thiazino[2,3-b]quinoxaline (1b) and 4a,12a-dihydro-12H-benzo[5,6][1,4]thiazino[2,3-b]quinoxaline (1c), 8-(isopropyl)-2-methyl-5-nitroquinoline, 8-(isopropyl)-2-methylquinolin-5-amine (4a) and have been deleted. We left the spectroscopic characteristics of compound previously described by our team namely (4a,12a-dihydro-12H-benzo[5,6][1,4]thiazino[2,3-b]quinoxalin-12-yl)(phenyl)methanone (1). In the present work, the characteristic is more detailed. The 2,8-dimethylquinoline was obtained through our new methodology. The 2,8-dimethyl-5-nitroquinoline and 2,8-dimethylquinolin-5-amine (4b) are new products. We elaborated a new procedure of synthesis of known: 2,8-dimethylquinoline, 3,7-dinitro-10H-phenothiazine 5-oxide (3). We left the spectroscopic characteristics of this compound to show that both procedures lead to the same molecule.

  1. Furthermore, as the paper refers to the spectroscopic characterization of phenazine and phenothiazine derivatives, it should also include photophysical properties of the phenazine derivatives (5, 6 and 7).

Answer: We completely agree with the reviewer that the title of the article implies the inclusion of spectroscopic characteristics of compounds 5, 6, and 7. Molecule 7 is a side product only. This is an azoaryl derivative. However, this article focuses on the characterization of selected compounds, phenazine and phenothiazine derivatives. A separate article will present an in-depth characterization of the new compounds; phenazine and phenothiazine derivatives.

Some improvements in the writing have been made. I have carefully revised the whole manuscript and tried to avoid grammar or syntax errors. Besides, I have asked several skilled authors of English language papers to check the English. Thank you so much for your help. I appreciate it.

Yours sincerely, Jacek Nycz (on behalf of all co-Authors)

Reviewer 3 Report

Interesting paper deserving publication.

The Authors synthesized new products from the phosphorylation reaction of 10H-phenothiazine. X-ray structures have been determined for several compounds. Among them compound 2a exhibits interesting structural features namely an anomeric effect. The influence of the sulfur and phosphorus atoms on the structural properties of 10H-phenothiazine has been investigated, in particular by comparison with phenazines. The absorption and emission properties of compounds have been investigated experimentally and theoretically using standard DFT and TDDFT computations. Original and interesting results have been obtained, among them the emission spectra of 2a, exhibiting both fluorescence and phosphorescence, which have been studied in detail.

Few points could deserve further developments, for instance the solvent role on the anomeric effect. Concerning the optical properties, since the MOs involved in the excitations are localized in the phenothiazine ring, one could expect a rather similar absorption spectrum for 2a (figure 8, figure S7) and phenothiazine (figure S8), and this not the case. Is there an explanation?

Minor point:

“Figure S9. Molecular orbitals for 2a calculated at the TDDFT/B3LYP/6-31G(d,p) level of theory.” Please, delete TDDFT/ from the title of the figure

Author Response

Dear Editor,

We want to thank the referees whose comments helped us improve the manuscript's clarity. Further, we provide detailed comments on the referees’ notes. All changes made in the text are highlighted in green color.

 A point-by-point response to comments from Reviewer 3

Interesting paper deserving publication.

The Authors synthesized new products from the phosphorylation reaction of 10H-phenothiazine. X-ray structures have been determined for several compounds. Among them compound 2a exhibits interesting structural features namely an anomeric effect. The influence of the sulfur and phosphorus atoms on the structural properties of 10H-phenothiazine has been investigated, in particular by comparison with phenazines. The absorption and emission properties of compounds have been investigated experimentally and theoretically using standard DFT and TDDFT computations. Original and interesting results have been obtained, among them the emission spectra of 2a, exhibiting both fluorescence and phosphorescence, which have been studied in detail.

Few points could deserve further developments, for instance the solvent role on the anomeric effect. Concerning the optical properties, since the MOs involved in the excitations are localized in the phenothiazine ring, one could expect a rather similar absorption spectrum for 2a (figure 8, figure S7) and phenothiazine (figure S8), and this not the case. Is there an explanation?

Answer: Many thanks for this important recommendation. The role of the solvent on the anomeric effect will be discussed in our further studies. For both molecules 2a and 3 the MOs involved in the excitations are localized in the phenothiazine ring; however, the involved MOs have different shapes and charge transfer (CT) nature (Figures 8b and 13b). For 2a, the non-bonding p oxygen orbitals interact with phosphorus, and the charge is transferred through the nitrogen atom to the thiazine-type ring. For 2a molecule, the stabilizing hyperconjugation effect of non-bonding 2p oxygen orbitals via phosphorus with antibonding p*-accepting orbitals of the thiazine type ring is strong. Moreover, the first band in the experimental spectrum of 2a corresponds to a series of the low-intensity transitions S0→S1, (HOMO→LUMO), S0→S2 (HOMO→LUMO+1), and S0→S3 (HOMO→LUMO+2, HOMO-1→LUMO).

For 3, there is no stabilizing hyperconjugation effect; the HOMO is antibonding orbital in respect to C-N and C-S bonds in the central ring showing there a strong C-C bonding character; whereas the LUMO is mostly non-bonding orbital in the center. For 3 the S0→S1 transition is provided by the HOMO→LUMO configuration and possesses a charge transfer (CT) nature; this is mainly CT from sulfur to nitro groups.

All these features can explain the different shapes of the absorption spectra for 2a and 3 and are discussed in the main manuscript.

Minor point:

“Figure S9. Molecular orbitals for 2a calculated at the TDDFT/B3LYP/6-31G(d,p) level of theory.” Please, delete TDDFT/ from the title of the figure

Answer: Many thanks for this comment. We have deleted “TDDFT/” from the title of the Figures 8 and 13 (main text), and Figures S7, S8, S9, and S10 (SI material)

Some improvements in the writing have been made. I have carefully revised the whole manuscript and tried avoiding grammar or syntax errors. Besides, I have asked several skilled authors of English language papers to check the English. Thank you so much for your help. I appreciate it.

Yours sincerely, Jacek Nycz (on behalf of all co-Authors)

Round 2

Reviewer 1 Report

I thank the authors for answering my questions and improving the manuscript.

The authors have taken into account all my comments.

However, I recommend moving the Scheme 5 with the compounds structure to part 2.1.

In part "4 Conclusions", the authors need not only to list, but also to summarize the obtained results. Please also add information on their possible practical application.

Author Response

A point-by-point response to comments from Reviewer 1

I thank the authors for answering my questions and improving the manuscript.

The authors have taken into account all my comments.

However, I recommend moving the Scheme 5 with the compounds structure to part 2.1.

Answer: The suggested correction has been made, the Scheme 5 is now Scheme 1, and the other Schemes in the right order.

In part "4 Conclusions", the authors need not only to list, but also to summarize the obtained results. Please also add information on their possible practical application.

Answer: The suggested correction has been made. Many thanks for this important crochet info. The new Text: “We described and analyzed the crystal structure of the N-benzoylated phenothiazine-phenazine hybrid molecule 1. Compounds of this type possess rich future chemistry potential for photoluminescent materials [27] and polymerization photoinitiating systems”, and “molecules 6a-d are structurally related to the well-known biological stain neutral red, which analogs exhibit promising photosensitizing [70] and electrochemical properties”.

Yours sincerely, Jacek Nycz (on behalf of all co-Authors)

Reviewer 2 Report

The paper is significantly improved and can be published in current form.

Author Response

A point-by-point response to comments from Reviewer 2

The paper is significantly improved and can be published in current form.

Answer: Many thanks for the recommendation of our work.

Yours sincerely, Jacek Nycz (on behalf of all co-Authors)